# Planning and Acting While the Clock Ticks

**Primary Keywords:** *(6) Temporal Planning*

## Abstract

Standard temporal planning assumes that planning takes place offline, and then execution starts at time 0. Recently, situated temporal planning was introduced, where planning starts at time 0, and execution occurs after planning terminates. Situated temporal planning reflects a more realistic scenario where time passes during planning. However, in situated temporal planning a complete plan must be generated before any action is executed. In some problems with time pressure, timing is too tight to complete planning before the first action must be executed. For example, an autonomous car that has a truck backing towards it should probably move out of the way now, and plan how to get to its destination later. In this paper, we propose a new problem setting, called concurrent planning and execution, in which actions can be dispatched (executed) before planning terminates. Unlike previous work on planning and execution, we must handle wall clock deadlines that affect action applicability and goal achievement (as in situated planning) while also supporting dispatching actions before a complete plan has been found. We extend previous work on metareasoning for situated temporal planning to develop an algorithm for this new setting. Our empirical evaluation shows that when there is strong time pressure, our approach outperforms situated temporal planning.

## 1 Introduction

Agents operating in the real world, such as robots, must be able to handle the fact that time passes in the real world. In temporal planning (Fox and Long 2003), the problem formulation accounts for time passing during plan *execution*. However, the problem formulation does not account for the time that passes during *planning* and thus is suitable for offline planning or for situations where planning time is insignificant compared to execution time.

Situated temporal planning (Cashmore et al. 2018) was proposed as a problem formulation in which time passage during planning is accounted for. In situated temporal planning, the planner must output a complete plan in a timely manner, that is, before it is too late to execute that plan. Situated temporal planning is useful when there are temporal constraints (such as deadlines), and when planning time might affect the feasibility of meeting such deadlines. For example, situated temporal planning was shown to be useful for online replanning for robots (Cashmore et al. 2019).

However, situated temporal planning is still constrained to output a *complete plan* before execution begins. This might be problematic in situations with tight deadlines, which do not allow enough time to find a full plan before the first action is executed. For example, consider a robot assigned with preparing dinner — a situation marked by a concrete deadline. Despite not having a finalized plan for procuring and assembling all the ingredients, the robot might in some situations be forced to initiate meal preparation, such as boiling water on the stove (de Pomaine 1930), to ensure timely completion and meet the dinner deadline.

Previous work on combining planning and execution, such as IxTeT-ExEc (Lemai and Ingrand 2004) and ROSPlan (Cashmore et al. 2015) (among others), focused on how to integrate an offline planner with a reactive executive to create an online planning and execution system. This does not address problems with tight deadlines, where plans can become infeasible during the planning process due to a deadline expiring during search. Other work addressed the question of when to commit to dispatching an action during search (Gu et al. 2022), but this work also does not reason about deadlines, and is thus inapplicable to such settings.

In this paper, we formalize the problem of *concurrent planning and execution* and describe an algorithm for solving it. Our algorithm builds on the state-of-the-art in situated temporal planning (Shperberg et al. 2021), extending it with the option to dispatch an action for execution during search, before a complete plan is found. Although the resulting system contains elements of a traditional executive, we still refer to it as a planner, albeit one that can dispatch actions.

The situated temporal planner we build upon employs a rational metareasoning approach (Russell and Wefald 1991) that tries to choose the best computational action based on the information currently available to the planner. Given the inherent complexity of the full metareasoning problem, the planner adopts a pragmatic approach by implementing decision rules derived from a simplified version of the metareasoning problem (Shperberg et al. 2019). Prior work has extended the abstract metareasoning problem to model concurrent planning and execution (CoPE) (Elboher et al. 2023).

Thus, it seems natural to employ the metareasoning approach for concurrent planning and execution in the planner, thereby deriving an algorithm that seamlessly integrates both aspects. Unfortunately, as we discuss later in this pa-

per, the CoPE metareasoning model relies on over-optimistic estimates of the probability that some branch of the search tree will ultimately succeed in reaching a solution. Since dispatching an action is irreversible, this can lead to failure.

Therefore, we present a new simplified metareasoning model, referred to as CoPEM (Concurrent Planning and Execution with Measurements). This model incorporates the understanding that the search process acquires valuable information, enhancing our estimation of the probability of success across various search branches. The new metareasoning model accounts for the value of this information. While CoPEM yields an intractable POMDP, it clarifies the notion of value of information for additional search effort. This allows us to approximate that value, aiding in the decision-making process of whether to execute actions immediately or await additional information.

Based on these insights and approximations, we present a decision rule for determining whether to search or dispatch an action. This decision rule is integrated into the situated temporal planner of Shperberg et al. (2021), yielding a dispatching planner for concurrent planning and execution. An empirical evaluation shows that this system outperforms situated temporal planning whenever time pressure is tight.

## 2  Problem Statement

We define *concurrent planning and execution* in a manner closely aligned with situated temporal planning (Cashmore et al. 2018): as propositional temporal planning with Timed Initial Literals (TIL) (Cresswell and Coddington 2003; Edelkamp and Hoffmann 2004). The sole distinction between concurrent planning and execution and situated temporal planning is the ability to execute an action before we have a complete plan, which is formalized by slightly different constraints on *when* the output is produced.

Formally, a concurrent planning and execution problem $\Pi$ is specified by a tuple $\Pi = \langle F, A, I, T, G \rangle$, where $F$ is a set of Boolean propositions that describe the state of the world, $A$ is a set of durative actions, with $a \in A$ composed of a duration, $dur(a) \in \mathbb{R}^{0+}$, start condition $cond_\vdash(a)$, invariant condition $cond_\leftrightarrow(a)$, and end condition $cond_\dashv(a)$, all of which are subsets of $F$. The effects are given by start effect $eff_\vdash(a)$ and end effect $eff_\dashv(a)$, both of which specify which propositions in $F$ become true (add effects), and which become false (delete effects). $I \subseteq F$ is the initial state, specifying exactly which propositions are true at time 0. $T$ is a set of timed initial literals (TIL). Each TIL $l = \langle time(l), lit(l) \rangle \in T$ consists of a time $time(l)$ and a literal $lit(l)$, which specifies which proposition in $F$ becomes true (or false) at time $time(l)$, and $G \subseteq F$ specifies the goal, that is, propositions we want to be true at the end of plan execution.

As in situated temporal planning, TILs are seen as temporal constraints in *absolute time since planning started*. However, unlike situated temporal planning, where we require generation of a full plan before execution begins, in concurrent planning and execution we allow our algorithm to dispatch an action, even before a complete plan is available.

Formally, our algorithm outputs a sequence of pairs $\langle a, t_a \rangle$, where $a \in A$ is an action and $t_a \in \mathbb{R}^{0+}$ is the time when action $a$ is to start. The first requirement is that this sequence of actions forms a valid solution for the planning problem $\Pi$, that is, that all conditions hold at their respective time points, and that the plan achieves the goal, exactly as in standard temporal planning.

In more detail, we define a valid schedule by viewing it as a set of instantaneous *happenings* (Fox and Long 2003) that occur when an action starts, when an action ends, and when a timed initial literal is triggered. For each pair $\langle a, t_a \rangle$ in $\sigma$, we have action $a$ starting at time $t_a$ (requiring $cond_\vdash(a)$ to hold a small amount of time $\epsilon$ before time $t_a$, and applying the effects $eff_\vdash(a)$ right at $t_a$, and ending at time $t_a + dur(a)$ (requiring $cond_\dashv(a)$ to hold $\epsilon$ before $t_a + dur(a)$, and applying the effects $eff_\dashv(a)$ at time $t_a + dur(a)$). For a TIL $l$ we have the effect specified by $lit(l)$ triggered at $time(l)$. Finally, we require the invariant condition $cond_\leftrightarrow(a)$ to hold over the open interval between $t_a$ and $t_a + dur(a)$, and the goal $G$ to hold after all happenings have occurred.

However, uniquely to our problem, we must also ensure our algorithm does not dispatch actions in the past. Thus, we annotate the output from our algorithm with the time each pair was output. That is, we treat the output as a sequence $(t_1, \langle a_1, t_{a_1} \rangle), \ldots, (t_n, \langle a_n, t_{a_n} \rangle)$, where $t_1 \le t_2 \le \ldots t_n$, indicating that the pair $\langle a_i, t_{a_i} \rangle$ was output at time $t_i$. The requirement is that $t_i \le t_{a_i}$—meaning our algorithm commits solely to dispatching actions in the future. We remark that there is never any theoretical benefit in committing to an action in the future, and thus typically we would expect to see $t_i = t_{a_i}$. However, practical considerations might interfere with this, as we discuss below. For comparison, in situated temporal planning, the requirement is $\forall i : t_n \le \min t_{a_i}$. This formulation generalizes cases where the planner outputs a complete plan at once, in which case $t_1 = \cdots = t_n$. If our algorithm is able to emit a sequence of actions that forms a valid plan, without violating the laws of space-time by dispatching actions in the past, we call it *successful*.

Consider the aforementioned example of a robot preparing dinner. In this scenario, a TIL $l$ is utilized to enforce that the meal must be ready before dinner time, $time(l)$. If the planning time is substantial, $t_n$ will approach $time(l)$. Without concurrent dispatching, all actions must be executed within the timeframe $t_n - time(l)$, which could be infeasible, given the time-consuming nature of cooking. However, in our concurrent setting, the sole requirement is for the final action to be executed after $t_n$, significantly enhancing the feasibility of the task. For example, it is beneficial to dispatch long actions (such as boiling a large pot of water) early, as these will no longer be constrained to fit within the time window between when planning ends and the deadline.

## 3  Prior Work

The situated temporal planner we build upon (Shperberg et al. 2021) uses the OPTIC planner (Benton, Coles, and Coles 2012). OPTIC applies heuristic forward search in the space of sequences of happenings (snap actions). Many other temporal planners use different types of search techniques, such as constraint propagation (Vidal and Geffner 2006), or compilation to SAT (Rankooh and Ghassem-Sani 2015). These planners were designed for offline planning.

However, when we want to consider dispatching an action, it is much easier to do so in the context of a current state of the world, and thus relying on forward search seems to be the most natural approach. Of course, it is possible to adapt other forward search planners (Gerevini, Saetti, and Serina 2003; Vidal 2004; Eyerich, Mattmüller, and Röger 2009) with the ability to dispatch actions. However, the planner we build upon has most of the machinery needed for metareasoning, which we describe next, thus making our job easier.

### 3.1 Abstract Metareasoning

Rational metareasoning (Russell and Wefald 1991) provides a way to choose among different computational actions. The decision problem that metareasoning addresses is called the meta-level decision problem, and it can be formalized as an MDP, or as a POMDP when we have partial information. Computational actions in our setting include expanding a search node or dispatching an action, making the meta-level decision problem rather complicated.

Shperberg et al. (2019) address part of this metareasoning problem (excepting action dispatching), by abstracting from the intricacies of the plan state representation and search process. They model the problem using $n$ processes, denoted $p_1, \ldots, p_n$, where each process is dedicated to searching for a plan (each process can be thought of as representing a search node on the open list). Each process is described by a probabilistic performance profile, modeled by a random variable (RV) indicating the probability of process $p_i$ terminating successfully given processing time $t$; $M_i$ denotes the Cumulative Distribution Function (CDF) of this RV.

Processing must terminate before a deadline, which may be unknown during planning, and is thus also assumed to be a random variable. The CDF of the deadline by which time process $p_i$ must terminate in order for its solution to be usable is denoted by $D_i$. Note that the deadline is with respect to 'wall clock' time (total time allocated to all processes), while $M_i$ counts 'CPU time' (time allocated to $p_i$).

Under the assumption that information about the true deadline and processing time of process $p_i$ is only available when that process terminates, the problem is to find an optimal policy for scheduling processing time for all the processes, so that the probability that some process $p_i$ will deliver a plan before its deadline is maximal. A slightly simplified version of this problem, when time is discrete (assumed to be integer-valued) is known as Simplified Allocating planning Effort when Actions Expire, or S(AE)$^2$ for short. Solving S(AE)$^2$ optimally was shown to be NP-hard, but if the deadlines are known (called KDS(AE)$^2$ , with KD standing for Known Deadlines), the problem can be solved optimally in pseudo-polynomial time by dynamic programming (Shperberg et al. 2019, 2021).

As even solving the simplified problem using the pseudo-polynomial algorithm is too expensive, previous work relies on a simplified decision scheme called Delay Damage Aware (DDA), which is based on ideas used in the optimal KDS(AE)$^2$. The DDA scheme relies on the log-probability of failure (LPF) of allocating $t$ consecutive units of computation time to process $i$, starting at time $t_b$, denoted $LPF_i(t, t_b)$. To compute the LPF, we first compute the prob-

ability that process $i$ finds a timely plan when allocated $t$ consecutive time units beginning at time $t_b$, which is:

$$s_i(t, t_b) = \sum_{t'=0}^{t} m_i(t') \cdot (1 - D_i(t' + t_b)) \qquad (1)$$

where $m_i(t) = M_i(t) - M_i(t - 1)$, i.e. the PMF of $M_i$. The choice to use the log of the probability of failure allows us to treat it like an additive utility function, thus we define $LPF_i(t, t_b) = log(1 - s_i(t, t_b))$.

The DDA scheme allocates chunks of $t_u$ computational time units (where $t_u$ is a hyperparameter), and the utility of a process $i$ is defined by the log-probability of failure of allocating computation time to process $i$ in the next chunk (starting at time $t_u$, with a discount factor of $\gamma$), minus the log-probability of failure of allocating time to process $i$ now (thus accounting for the urgency of the process). The amount of computation time to use in the utility calculation is chosen by the *most effective computation time* for process $i$ starting at time $t_b$, defined as $e_i(t_b) = \operatorname{argmin}_t \frac{LPF_i(t, t_b)}{t}$, that is, the time allocation is chosen by its marginal gains. Putting this all together, the DDA scheme allocates the next unit of computation time to the process $i$ with maximal

$$Q(i) = \frac{\gamma \cdot LPF_i(e_i(t_u), t_u)}{e_i(t_u)} - \frac{LPF_i(e_i(0), 0)}{e_i(0)} \qquad (2)$$

### 3.2 MR in Concurrent Planning & Execution

An abstract metareasoning model for Concurrent Planning and Execution, called CoPE, was presented by Elboher et al. (2023). The CoPE metareasoning model extends S(AE)$^2$ by assuming that each process $p_i$ has already computed a plan prefix $H_i$ consisting of some actions (for example, if $p_i$ represents a node on the open list, $H_i$ are the actions leading from the initial state to that node). One is allowed to start executing actions from some $H_i$ before planning terminates (and concurrently with planning), but doing so invalidates all processes that have initial actions inconsistent with the prefix of $H_i$ already executed. The requirement now is to have at least one still valid process $p_i$ complete its computation *and* execute its $H_i$ before some *induced deadline* $ID_i$ (the induced deadline can be computed from $D_i$ and the action durations, but we omit the details here for the sake of brevity). Note that a CoPE problem instance where all $H_i$ are empty is also an S(AE)$^2$ instance.

For the special case of CoPE where the induced deadlines are known (denoted KIDCoPE), it is possible to reduce the problem to multiple instances of $KDS(AE)^2$, which in turn can be solved in pseudo-polynomial time by dynamic programming (Elboher et al. 2023). This is done by choosing an execution time for all the actions in some $H_i$. The function which defines the execution time for each action in $H_i$ is called an *initiation function*, and we denote it by $f$.

The semantics is that each action $a \in H_i$ is executed beginning at $f(a)$, unless $a$ becomes redundant because a complete timely plan that does not use $a$ is found before $f(a)$. Recall that when action $a$ is executed, it invalidates any process $j$ that has an $H_j$ inconsistent with $a$. Thus, given $f$, one can define an *effective deadline* $d_i^{eff}$ for each process $i$ as the

minimum of $ID_i$ and the execution time $f(a)$ for the first action inconsistent with $H_i$. Using the effective deadlines, we get, rephrasing Theorem 6 from (Elboher et al. 2023):

**Theorem 1** *Given a CoPE problem instance I and an initiation function f, using the effective deadlines as computation deadlines (and subsequently ignoring the $H_i$) defines a $KDS(AE)^2$ instance $KDS(I, f)$. The optimal policy for I restricted to action execution according to f is equal to the optimal computation policy for $KDS(I, f)$.*

Solving the resulting $KDS(AE)^2$ instances for all possible initiation functions $f$ and picking the one with the highest success probability is an algorithm for optimal solution of the KIDCoPE instance. Obviously, the complexity of the above algorithm is exponential in $\max_i |H_i|$, but special cases exist where the number of possible $f$ is polynomial (Elboher et al. 2023). We use a similar technique below.

### 3.3 Metareasoning in a Planner

So far, we have discussed abstract metareasoning models. Integrating these into a planner is not trivial. We now explain briefly how the situated temporal planner we build upon (Shperberg et al. 2021) uses the DDA decision rule in practice.

First, there are several adaptations to the node expansion process itself, accounting for TILs that occur during planning and pruning nodes for which it is already too late to start executing (Cashmore et al. 2018). Second, to use the DDA decision rule, the planner must estimate $M_i$ and $D_i$, which are used to compute the LPF.

An admissible deadline for node $i$ can be found by building a Simple Temporal Problem (STP) for the plan prefix $H_i$, solving it to find the latest feasible timestamp $tmax$ of each step, and taking the minimum of these across all steps:

$$latest\_start(H_i) = \min_{a_j \in H_i} tmax(a_j)$$

In practice, a more informative but inadmissible estimate is found based on the temporal relaxed planning graph (TRPG) (Coles et al. 2010) heuristic by additionally including the relaxed plan $\pi_i$ in this STP. This estimate is called the *estimated latest start time* for node $i$ and is used as a known deadline (that is, $D_i$ assigns a probability of 1 to this being the deadline).

To estimate $M_i$, the planner relies on the notion of expansion delay (Dionne, Thayer, and Ruml 2011)—the average number of expansions between when a node is generated and when it is expanded. A distribution is built around this estimate based on statistics collected during search; the details of how this is done are omitted for the sake of brevity.

With these estimates of $M_i$ and $D_i$, the planner can compute $Q_i$ for each node on the open list and sort the open list based on $Q$. Since the DDA scheme is based on allocating $t_u$ units of computation time to the chosen process $i$, the planner performs $t_u$ expansions in the *subtree* rooted at $i$; after $t_u$ expansions, the non-expanded (frontier) nodes in this subtree are added to the open list and another state is chosen according to $Q$. Additionally, as the estimates for $M_i$ could change (because the statistics collected during search to estimate $M_i$ change), the $Q$ value is recomputed for all the nodes on the open list every $t_u$ expansions.

So far we discussed how DDA, a metareasoning scheme for $S(AE)^2$, was integrated into a planner. The issue with integrating CoPE into a planner is that the planner's estimates of the $M_i$ and $D_i$ distributions can be quite far off the mark, especially early in the search; thus might lead to wrong decisions. In situated temporal planning, this is not critical, because a wrong decision wastes only some search effort (a few node expansions). However, in concurrent planning and execution, a wrong decision to dispatch an action can frequently be fatal. Therefore, our first step is developing an improved metareasoning model, which accounts for the information gathered by 'measurements' during search.

## 4 Metareasoning with Measurement Model

We now present our new CoPEM abstract metareasoning model, extending CoPE by explicitly making the more realistic assumption that computation actions deliver information that can update the distributions. CoPEM is obviously at least as computationally hard to solve optimally as CoPE, so except for restricted cases, we do not attempt to solve it optimally. Its main role is to specify a notion of what it means to be optimal in concurrent planning and execution. Nevertheless, we leverage ideas from optimal solutions of the restricted cases towards an actual implementation solving the concurrent planning and execution problem from Section 2.

### 4.1 The CoPEM Model

As in the CoPE model, we have $n$ processes $p_1...p_n$, all searching for a plan starting at a known initial state. Each process $p_i$ has already computed an initial action sequence $H_i$, where each action in $H_i$ has a specific time window for execution. For each process we have a random variable $M_i$, a performance profile determining how long it needs to compute until termination. Random variable $D_i$ is a distribution over the induced deadline of process $p_i$: the time by which the last action in $H_i$ and the rest of any solution found by process $p_i$ must be executed in order to be successful. In general, the random variables may be dependent.

There are three types of actions: real-world actions to be dispatched corresponding to the next action from some $H_i$, a computation action $c_i$ allocating a processing time unit to process $p_i$, or commit to a complete correct plan found by any terminated process $p_i$. All actions are non-preemptible and mutually exclusive, except that computation actions can be run concurrently with at most one real-world action. A computation action $c_i$ can make the process terminate (with probability determined by $M_i$), in which case the process delivers a solution and its true deadline is revealed.

Up to now, this is the same as the CoPE model. However, in CoPEM the distributions over the random variables $M_i$ and $D_i$ are just priors: a computation $c_i$ also delivers an observation $o \in O_i$ (for some observation space $O_i$) as evidence that affects the posterior distributions over the random variables according to a known *measurement model*. In other words, the action $c_i$ has a stochastic effect on what the $M_i$ and $D_i$ distributions would be in the next (belief) state, after updating them according to the observation $o$.

In order to parameterize a restriction on the model complexity, we define a parameter $K$, the time at which we no

longer allow a real-world action to be dispatched 'early' (i.e. before the computation terminates), and a parameter $L$, the last time at which observations can be received (when a computation action $c_i$ performed at time $t$ causes the observation $o$ to be received in time to make the decision at time $t + 1$). The CoPE model is a special case of CoPEM where $L = 0$ and $K = \infty$.

We assume here that the measurement model is known and that we can perform Bayesian updating on the runtime and deadline distributions. Under this assumption, like many metareasoning problems, the CoPEM model is a POMDP, the solution of which is intractable: potentially exponential in the number of time units in the model, thus certainly not optimally solvable in real-time. We therefore examine some special cases, and leverage their solutions towards a greedy decision-making algorithm to handle CoPEM in practice.

### 4.2 Basic Tractable Case

We begin with the restricted CoPEM case which we call *fully resolved*. Given a CoPEM instance $I$ and a commitment to execute all the actions in some $H_i$ at certain times (an initiation function $f$), $(I, f)$ is fully resolved if the induced deadlines are known and no further information can be obtained by computations about any of the $M_i$ distributions.

**Theorem 2** *If $(I, f)$ is fully resolved, the optimal policy for $I$ under initiation function $f$ is equal to that of an equivalent $KDS(AE)^2$ instance $I(I, S)$.*

**Proof**: If no further information on the $M_i$ can be obtained by computation, and the induced deadlines are known, we have an instance of KIDCoPE. Since we also have a fixed initiation function $f$, by Theorem 1 we can construct an instance $KDS(I, f)$ whose optimal computation policy is also optimal for $I$. $\square$

Leveraging this tractability property to solve instances of CoPEM would be advantageous. However, its application is nontrivial, since a CoPEM instance is not fully resolved, even for $K = L = 0$ and with known induced deadlines, since any of the $H_i$ actions could be dispatched at $t = 0$, i.e. we do not have an initiation function.

Nevertheless, consider a CoPEM instance restricted to $K = L = 0$, known induced deadlines, and $|H_i| = 1$ for all $i$. Opting not to dispatch an action at $t = 0$, we are not allowed to dispatch any actions early subsequently. Since no further information on $M_i$ can be acquired, effectively we have a fully resolved equivalent instance. Consequently, we can formulate an equivalent $KDS(AE)^2$ instance denoted as $\mathcal{I}_0$. Likewise, when opting to dispatch the only available action $a \in H_i$ for some $1 \leq i \leq n$, $K = 0$ implies $f(a) = 0$, so again we have a fully resolved instance, We denote the respective $KDS(AE)^2$ instance by $\mathcal{I}_i$.

Once we generate these $n + 1$ $KDS(AE)^2$ instances, we solve each of them optimally in pseudo-polynomial time by dynamic programming, obtaining a value $v_i$ (probability of success) for each of them. Our dispatch decision is then given by $\arg\max_{i=0}^{n} v_i$, where a decision of $i > 0$ indicates dispatching the first action in $H_i$, and a decision of $i = 0$ indicates not dispatching any action. This scheme, henceforth called Solve00($\cdot$), returns the best $v_i$ and (optionally) the solution for the respective $KDS(AE)^2$ instance.

**Theorem 3** *Given an instance $I$ of CoPEM with $K = L = 0$, known induced deadlines, and $|H_i| \leq 1$ for all $1 \leq i \leq n$, Solve00($I$) yields the optimal policy for $I$.*

Note that the above method is applicable for $|H_i| > 1$, but the DP solution to the $KDS(AE)^2$ instance is not guaranteed to be an optimal CoPEM solution beyond $|H_i| = 1$.

### 4.3 Extension to Incorporate Measurements

The above restrictions can be relaxed to scenarios with a bounded number $K$ of informative (w.r.t. $M_i$) computation actions and $L$ dispatch decisions, as long as our observation space $O_i$ for each computation action $c_i$ is finite. Let $O_{max} = \max_i |O_i|$. We focus on $K = L = 1$, which is similar in spirit to the Russell and Wefald myopic assumption (Russell and Wefald 1991; Tolpin and Shimony 2012).

We also assume here known induced deadlines, and jointly independent $M_i$ distributions. For clarity, we further assume that a computation action $c_i$ yields perfect information, i.e. an observation $o$ equal exactly to the remaining computation time for process $p_i$. Note that this assumption can be easily adjusted should an alternative measurement model for each computation action be available.

With these assumptions, the only uncertainty is in the $M_i$, and in observations to be received after the first computation action. Perfect information implies that computation $c_i$ will observe $o = t$ with probability $P(o = t) = m_i(t)$.

For conciseness and concreteness, we define notation for CoPEM and $KDS(AE)^2$ instances created due to dispatching decisions and/or observations, and how they are defined. Let $\mathcal{I}$ be the original CoPEM instance with $K = 1, L = 1$. We break this down into two subcases:

**Not dispatching at time 0:** Executing a computation $c_i$, we then get an observation $o \in O_i$, and in each such case we need to optimize a new CoPEM instance, where the current time is $t = 1$. We can shift the origin of the $t$ axis to 1 and treat the resulting state as a $K = L = 0$ instance. Let us denote each such problem instance by $\mathcal{I}(c_i, o)$. Denote by subscripting $\mathcal{I}_i$ the problem instance resulting from dispatching decision $i$, which is dispatching the (first and only) action in $H_i$, or deciding not to dispatch at this time for $i = 0$. An instance $\mathcal{I}(c_j, o)$ after a dispatch decision, denoted $\mathcal{I}_i(c_j, o)$ is now a $KDS(AE)^2$ instance, since no more observations can be received, and no additional early dispatch decisions are allowed. Note that the distribution model of $\mathcal{I}_i(c_j, o)$ is the same as that of $\mathcal{I}_i$ for all processes, except for $m_j$ which is replaced by the degenerate distribution $\delta(o)$ (where $\delta$ is the delta function).

**Dispatching at time 0:** Alternatively, we can decide on an early dispatch in $\mathcal{I}$ at time $t = 0$, resulting in an instance $\mathcal{I}_i$. Note that the result is not a $KDS(AE)^2$ instance. However, since no further actions can be dispatched, each $\mathcal{I}_i$ is essentially a $K = 0, L = 1$ instance. But after the next computation $c_j$ is done, we receive the observation $o$ and get a $KDS(AE)^2$ instance again: not the same as $\mathcal{I}_i(c_j, o)$ because here an action from $H_i$ has been dispatched at time 0, rather

than 1, so we denote this by $\mathcal{I}_i^0(c_j, o)$. To find the probability of success of the optimal policy, compute:

$$Solve01(\mathcal{I}_i) = \max_j \sum_{o \in support(m_j)} m_j(o) Solve00(\mathcal{I}_i^0(c_j, o)) \quad (3)$$

and any $c_j$ achieving the maximum is optimal.

All in all, the $K = L = 1$ case is handled by pseudo-polynomial time computation of the following equations, denoted as Solve11. For all $1 \le i \le n$:

$$P_i = \sum_{o \in support(m_i)} m_i(o) Solve00(\mathcal{I}_i(c_i, o)) \quad (4)$$

$$P_i' = Solve01(\mathcal{I}_i) \quad (5)$$

Then, select the policy corresponding to the maximum value of all $P_i$ (compute at time 0 policies) and all $P_i'$ (dispatch at time 0 policies). Essentially we evaluate a depth 2 expectimax tree, with leaf nodes being KDS(AE)$^2$ instances, and return the best policy and its corresponding probability of success. The overall complexity of this scheme is $O(T^2 n^4 O_{max})$ where $T$ is the number of time steps. $T^2 n^2$ is the time taken to solve a single KDS(AE)$^2$ instance by dynamic programming, and $n^2 O_{max}$ simply counts the loops to compute the equations.

In principle, this scheme can be extended to greater $K$ and $L$, paying a factor $n^{L+K} O_{max}^L$ by optimizing over all possible action and observation sequences of length $L$ and dispatching decision sequences of length $K$. However, since even the $K = L = 1$ case, despite being tractable, is too heavy to use in metareasoning, this is not examined here.

### 4.4 Leveraging the Tractable Case in Practice

In applying the CoPEM model to concurrent planning and execution, obviously our metareasoning assumptions of independence, known induced deadlines, and the myopic $K = 1, L = 1$ assumptions do not hold. Nevertheless we can use this as a first approximation. Additionally, even though the optimal algorithm for $L = 1, K = 1$ above is a (pseudo) polynomial time algorithm, it is far too slow to make metareasoning decisions guiding a heuristic search algorithm.

Let $P_0'$ be the success probability of instance $\mathcal{I}$ deciding not to dispatch any action, and under the assumption that computations do not provide information about any $M_i$. Examining the algorithm for $K = L = 1$, we see that an action $a$ in $H_i$ is potentially dispatched only if it has a probability of success $P_i'$ higher than $P_0'$ (not dispatching an action). The difference $P_i' - P_0'$ in probability of success is the gain for dispatching $a$. Alternately, one can do a computation $c_i$ first, and then decide on dispatching. This is worthwhile only if the expected utility (measured in probability of success $P_i$) is increased on average vs. dispatching an action immediately by doing $c_i$ first, and the gain $P_i - \max_j P_j'$ is called the net value of information (VOI) for $c_i$ (at the initial state). In an optimal policy, an action $a$ is dispatched immediately only if no $c_i$ has a positive net VOI.

A practical algorithm based on approximating the optimal policy would thus consider whether dispatching some action $a_i$ is beneficial (improves success probability), and given such an action, checks whether some computational action $c_i$ has a positive net VOI, in which case $a_i$ is not dispatched immediately. That is the gist of our proposed scheme.

Several complications hinder the implementation of this scheme. First, the success probability values are computed exactly in KDS(AE)$^2$ instances, but in fact the deadlines are also uncertain and distributions are not independent, making the success probability computed for a KDS(AE)$^2$ instance an incorrect estimate of the actual success probability, as well as too slow to use for metareasoning within a planner. We also lack a good measurement model for the computations, and the perfect information assumption is also not realistic. Overcoming these problems towards an approximate realization of this scheme within a planner is described next.

## 5 Implementation within the Planner

The insight from the optimal solution of the restricted case is that we need to measure the expected gain for dispatching an action, and to consider VOI of computation. Below we discuss how these are actually done in the planner.

### 5.1 State-Space Modifications to Support Acting

We assume that at any given point we have dispatched $m$ plan steps $\pi = [a_1 .. a_m]$ (where $m$ is initially 0), with $dispatch\_time[j]$ being the time at which step $j \in [1..m]$ is to be executed. All states on the open list begin with these $m$ steps, so the resulting planning task is equivalent to search from the state reached by those $m$ steps; subject to states respecting temporal side-constraints as described next.

First, in the situated temporal planners, the STP used to capture the temporal constraints on a plan (Cashmore et al. 2018), included the temporal constraints required in OPTIC, and also required $t(a_j) \ge t_{now}$ for all plan steps – because execution cannot start earlier than $t_{now}$ (the current time). In our case, as the first $m$ actions have been dispatched – so can go 'before now' – we keep the temporal constraints required in OPTIC, but instead also require:

$$\forall a_j \in H_i \quad \begin{matrix} t(a_j) = dispatch\_time[j] & \text{if } j \le m \\ t(a_j) \ge t_{now} & \text{otherwise} \end{matrix}$$

Second, an STP is additionally used to find the $dispatch\_time$ values themselves. If at time $t_{now}$ the decision is made that the $m + 1$th plan step to be dispatched is the snap action $a_{m+1}$, we take the STP that would be built for the state reached by the snap action sequence $[a_1 .. a_{m+1}]$, and set $dispatch\_time[m + 1]$ to the earliest feasible value $tmin(a_{m+1})$ of step $a_{m+1}$ in this STP: the earliest time it can occur considering the ordering constraints between plan steps, and the dispatch times of the previous steps.

Third, we redefine the notion of the 'latest start time' for states. The scalar value described in Section 3.3 is defined with respect to all plan steps in an STP. As this would now include the $m$ steps that have been dispatched, we are instead interested in a latest start time that is conditional on $m$, i.e. what is the latest time we must dispatch step $m + 1$:

$$latest\_start(H_i, m) = \min_{a_j \in H_i | j > m} tmax(a_j)$$

Finally, we must consider the consequences of action dispatch on the open list, and on duplicate state detection (which in OPTIC and prior work is through maintaining a set of memoized states). The open list issue is easy: if we dispatch the snap action $a_m$ as step $m$, then we remove from the open list every state whose plan prefix $H_i$ does not have $a_m$ as step $m$. For duplicate detection, we must be more careful:

- We identify a set of states to 'un-memoize': any memoized state whose plan prefix $H_i$ does not begin with $[a_1..a_{m+1}]$, and remove them from the memoized states.
- We add to the open list for re-expansion any state $S_i$ expanded earlier in search, whose plan prefix $H_i$ does begin with $[a_1..a_{m+1}]$, and for which one or more of its successor states was pruned due to being equivalent to one of these un-memoized states.

## 5.2 Dispatch Estimates during Search

Having discussed modifications to the state space, we now turn to how to make metareasoning decisions in the planner. If we have dispatched the plan steps $\pi = [a_1..a_m]$, then our metareasoning decision is either to not dispatch something yet, or to dispatch *now* one of the snap-actions $next = [\alpha_1..\alpha_n]$ applicable in the state reached by $\pi$. We need to assess the utility of each of these possibilities, i.e. the probability of success in each case, which we assume to be related to its LPF via $1 - e^{LPF}$. We denote the probability of success for not dispatching, and for each of $next$, as $P_{Nd}$ and $P_{d1}..P_{dn}$, respectively.

As these utilities are unknown, we approximate them by simulating for each case what would be the allocations of the computation time to processes in the ensuing search, over a simulated open list $sim\_open$. When estimating $P_{Nd}$, we use $open$: the open list at the current moment in search. For $P_k \in [P_{d1}..P_{dn}]$, we consider only the nodes on the open list whose plan step $m + 1$ is $\alpha_k$:

$$sim\_open(open, \alpha_k) = [H_i \in open \mid H_i[m+1] = \alpha_k]$$

We then compute context-dependent LPFs for nodes on the simulated open list, with a context $c$ being the number of dispatched steps: $c = m$ for $P_{Nd}$, $c = m + 1$ otherwise. This is to use the appropriate $latest\_start$ estimates: when calculating $LPF_i^c$ we use the same calculation as Eq 1, except we base $D_i$ on $latest\_start(H_i, c)$. Thus, $c = m + 1$ means the dispatch options benefit from step $m + 1$ having been dispatched. We approximate the LPF of allocating time to processes $p_1...p_n$ in this order, starting at time $t_b$, with $t_b = t_{now}$ for each of $P_{d1}...P_{dn}$ (we would dispatch *now*); and $t_b = t_{now} + t_{wait}$ for $P_{Nd}$, as 'not dispatching' means waiting some amount of time. We assume $t_{wait} = t_u$. Then, the LPF under context $c$ of an open list is:

$$LPF^c(p_1..p_n) = \sum_{i \in [1..n]} LPF_i^c\big(\mathbb{E}(M_i), t_b + \sum_{j \in [1..i]} \mathbb{E}(M_j)\big)$$

This is an imprecise (but empirically informative) measure in a number of important regards. First, we assume each node $p_i$ is allocated $e_i$ expansions – the expected number of expansions for it to reach the goal. This is reasonable if $p_1$ is the best option for reaching the goals, $p_2$ is the second-best

option, and so on; which in reality, would require *inter alia* a perfect heuristic. Second, the order in which processes are considered is crucial: we fix the order here according to a snapshot of $open$, whereas in search proper, the order is due to the $Q$ values of processes, which in turn are a function of the time at which computation is to be allocated to them.

## 5.3 Metareasoning for Action Dispatch Decisions

Having defined how we estimate LPFs for candidate options (not dispatching yet, or dispatching some action) we now formulate dispatch rules. A naïve rule would be 'dispatch the best $\alpha_k$ where $P_{dk} > P_{Nd}$'. Since we only have inherently noisy and possibly biased estimates (due to expansion delay, a global one-step-error estimate, and an imperfect heuristic), rather than actual probabilities, we must be wary of dispatching an action if the benefit of doing so is small, and/or little search has been performed in the subtree reached by $\alpha_k$ in order to substantiate its probability approximation.

To address these issues, we first identify dispatch candidates $explored \subseteq next$, where $\alpha_k \in explored$ if a minimum number of expansions have been performed in its subtree, and $sim\_open(open, \alpha_k)$ is of a minimum size (these are hyperparameters). Then, if $explored \neq \emptyset$ we find:

$$\alpha_x = \underset{\alpha_k \in explored}{\arg\min} \; P_{dk}$$

and dispatch $\alpha_x$ if $P_{dx} > P_{Nd} + dt$ where $dt$ is our *dispatch threshold*. A caveat of this, however, is that if search is predominantly focusing on the subtree beneath only a subset $next$ (which is typically the case for heuristic search), any $\alpha_k \notin explored$ will never be considered for expansion, even if their probabilities of success are ostensibly better. To address this, we additionally find:

$$\alpha_y = \underset{\alpha_k \in (next \setminus unexplored)}{\arg\min} \; P_{dk}$$

and if $P_{dy} > P_{Nd} + sft$, where $sft$ is our *subtree focus threshold*, we constrain search to only expand nodes beneath $\alpha_y$. This embodies common sense, insofar as while we do not want to dispatch an action with nebulous evidence, there is an intuitive value of information argument.

# 6 Empirical Evaluation

Concurrent planning and execution is meant to be used in situations with time pressure. Unfortunately, most International Planning Competition domains were designed for offline planning and do not have inherent time pressure. Therefore, in our evaluation we focus on domains inspired by real problems involving robots: planning problems from the Robocup Logistics League (RCLL) planning competition (Niemueller, Lakemeyer, and Ferrein 2015) with one, two, and three robots, as well as planning problems for a Turtlebot performing an office delivery task. These were both used to evaluate situated temporal planning (Shperberg et al. 2021). The results for Turtlebot are not interesting – nearly every instance is solved by every configuration, and thus these results are relegated to the supplementary material.

We compare our concurrent planning and execution approach, denoted by *disp*, to the situated temporal planner

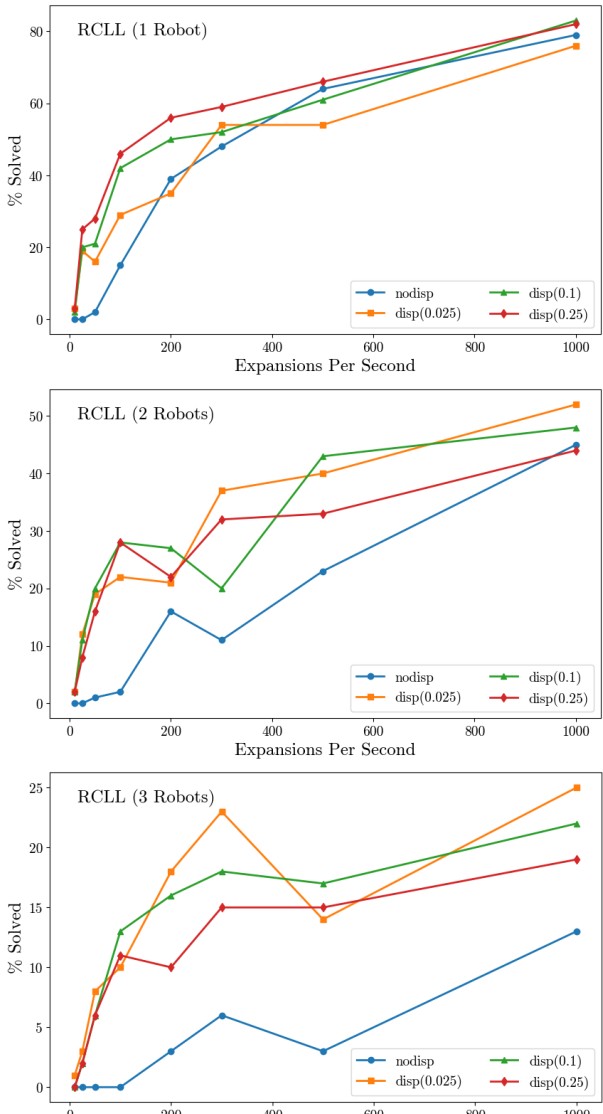

Figure 1: Problems solved vs. expansions-per-sec: RCLL

theoretical results.

Furthermore, because both approaches rely on the same planner, we choose to measure planning time by the number of expanded nodes rather than real wall-clock time. This allows for reducing random noise due to timing issues, and makes all of our experiments deterministic. Specifically, we use a user-specified parameter specifying how many node expansions we can perform per second, and then timing in the planner is based on the number of nodes expanded so far (divided by this parameter) rather than on the time that actually elapsed. We can then vary the number of expansions per second, simulating different 'CPU speeds'.

Figure 1 shows the number of problems solved by both approaches for different CPU speeds. As the results show, when the CPU speed is low (meaning fewer nodes can be expanded before the deadline expires), the benefit from using concurrent planning and execution is high. When the CPU is fast enough, the difference decreases. Furthermore, in cases where the CPU exhibits sufficient speed to solve the problem before the deadline, it is plausible that the no-dispatch strategy could surpass all dispatching policies, which may commit to a suboptimal action. This is in line with the theoretical limit of an infinitely fast CPU, where the best approach is to use offline planning to find a complete plan, and then start executing it at time 0. On the other hand, CPU speeds of robotic space explorers are typically at least an order of magnitude less than CPU speeds of computers on Earth, and thus we believe this is one area where our approach can be very useful.

## 7  Discussion and Future Work

We have formalized the CoPEM metareasoning problem, aimed at bridging the gap between planning and acting concurrently. Insights from optimal solutions of a restricted version of CoPEM are used in a new metareasoning algorithm deciding whether to dispatch an action during search. Despite having only rudimentary probability estimates and no realistic measurement model (improving them is an important issue for future work) our empirical evaluation shows that our algorithm works well under time pressure. In future work, we intend to develop a way to identify whether a current problem instance has strong time pressure, thus automatically switching between concurrent planning and execution, situated planning, and offline planning.

We are still a long way from the integration of our algorithm with real robots. First, there are technical challenges involved with actually dispatching an action on real hardware. Second, the real world features uncertainty. Actions can fail, or take longer or shorter than expected to execute, and these must be handled by the executive. One possibility is to leverage the replanning compilation suggested for situated temporal planning (Cashmore et al. 2019), which can also be used for our concurrent planning and execution formulation. Additionally, uncertain action durations can be handled as part of the planning process (Cimatti et al. 2018). Thus, we believe the framework presented here can serve as a principled basis for an executive that can be used on real robots.

(Shperberg et al. 2021) using its default parameters, denoted by *nodisp*. As both approaches use the same situated temporal planner, both are equally informed, thus allowing for a cleaner evaluation of the value of allowing dispatching actions before search completes, as well as the ability of our metareasoning approach to make these decisions correctly. Most parameters for *disp* are the same default parameters used for *nodisp*. For the new parameters introduced for making dispatching decisions, we set the minimum number of expanded nodes in the subtree for dispatching, as well as the minimum number of nodes in $sim\_open$ for expansion to 10. We vary the dispatch threshold (trying 0.25, 0.1, and 0.025), and set the subtree focus threshold to half the dispatch threshold. Ablation studies with other parameter settings (described in the supplementary material) show that these settings do capture a VOI criterion, which is important to the performance of our planner, as suggested by our

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
