# OpenReview forum: "Planning and Acting While the Clock Ticks"
_icaps-conference.org/ICAPS/2024/Conference — ICAPS 2024_

### Official Review · Reviewer_yLv2 · 2024-01-16

**Significance And Importance:** 1
**Soundness:** 3
**Novelty:** 1
**Clarity:** 3
**Overall Evaluation:** 1
**Confidence:** 4

**Weaknesses:**

0: Minor weaknesses requiring some work to be addressed for the paper to be accepted.

**Contributions Of The Paper:**

The paper presents an advancement on a planning and execution approach leveraging metareasoning. A series of work have been published considering more and more complex settings for guaranteeing viable planning and execution. The paper provides a comprehensive description of the general approach as well as the previous results. Then, it provides a specialised solution considering deadlines and dispatching actions. Namely, the paper describes a planner capable of dispatching actions even before a complete plan is generated. An empirical evaluation is provided to support the contribution.

**Ethical Considerations:**

(5) Excellent: The paper comprehensively addresses all of the applicable ethical considerations

**Nomination For Best Paper:**

No

**Questions For Authors:**

Is the proposed approach capable of dealing with dispatchability issues as defined by Tsamardinos et al?

Why are comparisons not considering other planning and execution approaches?

How hard is setting an internal model for supporting the proposed approach?

**Reproducibility:**

3: Authors describe the implementation and domains in sufficient detail.

**Strengths Of The Paper:**

The paper is well written. The proposed approach is built upon a strong formal framework and suitable details are provided to support the proposed solution. The empirical evaluation shows that the proposed approach dominates previous solutions considering different settings.

**Weaknesses Of The Paper:**

This paper represents the last of a rather long series of work that are investigating the planning and execution problem for temporal planning problems. The proposed advancements are rather limited. The experiments consider a comparison with previous (limited) solutions showing more efficient solutions and larger problems coverage. A comparison with other approaches is not considered.

Sot, the main weak point seems to be the limited significance of the contribution.

---

> ### Author Rebuttal · Authors · 2024-01-25
>
> We want to thank the reviewer for the helpful and detailed feedback. We will take all comments into account when preparing the final version, but address only the main points and questions here.
>
>
> Q1: In this paper, we are limiting our attention to actions with fixed durations, so dispatchability is not an issue. Indeed, if we consider actions with controllable durations, dispatchability issues might arise - we will address this in future work.
>
> Q2: We are unaware of previous work that can handle concurrent planning and execution in problems with deadlines against which we could compare, but we would certainly welcome any pointers that the reviewer might have.  Planners that do not execute carefully-chosen actions before planning completes will fail in these problems.
>
> Q3: We are not sure what the reviewer means by internal model.
> If this refers to the PDDL - this is part of the input to our planner.
> If this refers to the M_i and D_i distributions - these are estimated online, as described in Section 3.3.

---

### Official Review · Reviewer_5zZx · 2024-01-19

**Significance And Importance:** 2
**Soundness:** 3
**Novelty:** 2
**Clarity:** 3
**Overall Evaluation:** 1
**Confidence:** 4

**Weaknesses:**

1: Minor weaknesses that are easily fixable.

**Contributions Of The Paper:**

The paper introduces the concept of "concurrent planning and execution" as an extension to situated planning, allowing actions to be executed before planning concludes. Due to over-optimistic estimates in the CoPE approach, a new model, CoPEM, is proposed. The concurrent planning and execution problem is formally defined, incorporating rational metareasoning and its application in automated planning using CoPE. The CoPEM model leverages the assumption that actions provide information (observations) for updating distributions. Implementation details in the OPTIC planner are outlined. An empirical evaluation demonstrates the effectiveness of the proposed approach compared to three dispatch thresholds and a situated temporal planner, especially in low CPU speed scenarios. The paper concludes with a discussion on future extensions.

**Ethical Considerations:**

(1) Not Applicable: The paper does not have any ethical considerations to address

**Nomination For Best Paper:**

No

**Questions For Authors:**

I have a curiosity regarding the distinction between the proposed executor and a standard heuristic. In essence, a forward-search-based planner, equipped with effective heuristics, is expected to discern the correct action to prevent backtracking. In the execution phase, this equates to avoiding erroneously chosen actions.
An intuitive consideration suggests that the proposed approach introduces an overhead. It would be intriguing to conduct a comparison of resolution times between the proposed approach and those of a state-of-the-art planner.
It could also be interesting to compare a forward search planner with the proposed approach, executing the actions during the planning phase, to understand how many wrong choices are made bu its heuristics.

As per my understanding, the environment is assumed to be deterministic, with the exception of variations introduced by calculation times. However, the described observations may encompass information gathered from the environment through sensors. This inclusion of sensor-derived data could influence the search process contextually, especially in non-deterministic environments where, for instance, a robot operates alongside other agents.

Finally, it would be interesting to understand which PDDL fragment is used. For example, are competing durative actions allowed? Referring, for example, to the results discussed in the paper William Cushing, Subbarao Kambhampati, Mausam, and Daniel S. Weld. 2007. When is temporal planning really temporal? In Proceedings of the 20th international joint conference on Artificial intelligence (IJCAI'07). Morgan Kaufmann Publishers Inc., San Francisco, CA, USA, 1852–1859.

**Reproducibility:**

3: Authors describe the implementation and domains in sufficient detail.

**Strengths Of The Paper:**

The concepts proposed in the paper appear to align with dual processing theories, suggesting a parallel cognitive processing model. Somehow related, and possibly worth a comparison (although it does not follow a metareasoning approach), there is the paper by Booch, G., Fabiano, F., Horesh, L., Kate, K., Lenchner, J., Linck, N., Loreggia, A., Murgesan, K., Mattei, N., Rossi, F., & Srivastava, B. (2021). Thinking Fast and Slow in AI. Proceedings of the AAAI Conference on Artificial Intelligence, 35(17), 15042-15046.

**Weaknesses Of The Paper:**

The paper's content is dense and may benefit from figures to enhance reader comprehension. Currently, only one figure illustrating experimental results is provided.

It is not clear to me why, in the experimental results, there is no comparison with the previous CoPE.
Perhaps a performance comparison could help the reader better understand the differences between the two approaches.

---

> ### Author Rebuttal · Authors · 2024-01-25
>
> We want to thank the reviewer for the helpful and detailed feedback. We will take all comments into account when preparing the final version, but address only the main points and questions here.
>
> There is no comparison with the previous CoPE paper, since they never integrated CoPE with a planner. Initially (as mentioned around lines 88 and 342) we tried implementing CoPE directly in our planner, but quickly realized it was over-optimistic in dispatching actions and therefore we developed the CoPEM model.  We can certainly add these dismal results to the appendix for the final version.
>
> Q1: Comparing to a standard heuristic search planner is difficult, as they do not make clear when it might be good to dispatch an action during planning. One could compare to simply immediately executing the best action among the successors of the current state, but this could easily go down dead-ends. Approaches like Real-Time A* do something more reasonable, assuming a fixed time limit per move.
> The advantage of our approach is that it considers the time pressure for each move given the current deadlines, allocating more time for search when possible, and dispatching actions quickly when the time pressure is tight.
>
> Q2: This is an excellent observation. Indeed, we are considering extending our approach to non-deterministic environments and action durations in future work.
>
> Q3: We are solving problems formulated in PDDL 2.2, including required concurrency.

---

### Official Review · Reviewer_UwrV · 2024-01-22

**Significance And Importance:** 2
**Soundness:** 3
**Novelty:** 2
**Clarity:** 3
**Overall Evaluation:** 1
**Confidence:** 3

**Weaknesses:**

1: Minor weaknesses that are easily fixable.

**Contributions Of The Paper:**

The paper introduces an extension of the Concurrent Planning and Execution (CoPE) meta-reasoning model, recently presented in [Elboher et al. 2023]. As the original CoPE model was introduced to allow dispatching activities while planning so as to allow the planner to keep up with stringent temporal constraints, the extended model version presented in this paper - called Concurrent Planning and Execution with Measurements (CoPEM)- intends to further improve CoPE's performance by executing computation actions integrated within the search process, and exploiting the information stemming from post-computation observations. After presenting all the necessary background information to contextualize the reader, the paper describes the novel technical contributions to the model relative to the incorporation of the measurements within the used planner, analyzing some basic cases that make the resulting planning problem computationally tractable. After explaining how to make meta-reasoning decisions within the planner and how to formulate the concrete dispatching rules, an experimental session is presented where the CoPEM-based dispatching approach is compared against the same planner that uses no informed dispatching rule. The superiority of the presented approach is demonstrated over a known planning domain.

**Ethical Considerations:**

(1) Not Applicable: The paper does not have any ethical considerations to address

**Nomination For Best Paper:**

No

**Questions For Authors:**

1) Section 4.2, 1st column: it is unclear why the authors say that any of the actions in H_i could be dispatched at t=0. Aren't the actions contained in H_i temporally ordered? Isn't H_i defined as an action sequence?

2) Page 5, 1st column, "(i.e. before the computation terminates)": What is intended here by "computation"? The whole planning process or just the c_i computation action? Clarity on this point is of great importance.

3) Page 5, 2nd column, Section 4.3, line 2, 3: are the K and L variables fully changing their meaning w.r.t. Section 4.1?

4) Page 5, 2nd column, Section 4.3: "and in observations to be received after the first computation": this is unclear. How can there be further observations beyond the first computation if the maximum number of allowed computations is K = 1?

**Reproducibility:**

1: Difficult to reproduce because of missing detail.

**Strengths Of The Paper:**

Due to its density, the paper is a rather demanding reading. However, the topic is very interesting, and this makes the reading easier. Despite the heavy formalism, the structure of the paper is clear, and the authors did a good job in orderly unrolling all the necessary information. Also, despite this work is heavily based on previous literature work, the authors succeed in identifying a sufficiently interesting "advancement spot" to base their novel contribution on. To the best of my understanding, the paper is technically sound and contains no flaws. Indeed, the results obtained in the experimental section prove that the authors' idea to adapt the meta-reasoning approach to the concurrent planning and execution setting has succeeded.

**Weaknesses Of The Paper:**

Despite the theory behind the approach is clear and properly formulated, the solution proposed through the CoPEM model seems to be still at a rather premature level. The authors seem to be forced to resort to a significant number of simplifications in their analysis in order to reach a tractable solution. This introduces some doubts on the true applicability potential of the proposed solution in real-world environments, beyond the problems highlighted by the authors in the Discussion Section (i.e., domain uncertainty).
Moreover, the results show that the advantages obtained with the proposed method tend to promptly decrease as the CPU gets faster. Also, it would have been interesting to see also a comparison between the CoPEM and the original CoPE method, i.e., the Concurrent Planning and Execution approach that uses no measurement.

---

> ### Author Rebuttal · Authors · 2024-01-25
>
> We want to thank the reviewer for the helpful and detailed feedback. We will take all comments into account when preparing the final version, but address only the main points and questions here.
>
>
> Indeed, as you note, to achieve a tractable metareasoning decision rule, we make many simplifying assumptions. This is standard in metareasoning (since at least the days of Russell and Wefald), as solving the original meta-problem is computationally challenging.  We still feel that the theoretical work is valuable, as it provides motivation and indirect justification for the practical approximations.
>
> Regarding true applicability potential, we want to be clear that we are solving complex planning problems in a general-purpose PDDL planner, not just abstract metareasoning problems.  While we have not run our planner on a physical robot, we do feel that it is of practical utility.
>
> Certainly, as you note (and the paper explains, line 691), as CPU speed increases, there is little advantage to metareasoning or even concurrent planning and execution, as offline planning becomes instantaneous.  (We look forward to that day!)
>
> Q1: Indeed, only the first action in H_i can be dispatched at time t=0. The "any" refers to any $i$ - we will clarify this in the final version.
>
> Q2: "The computation" refers to the entire planning process - we will clarify this in the final version.
>
> Q3: Indeed, this a typo - the meaning of $K$ and $L$ is the same as in 4.1. We will fix this in the final version.
>
> Q4: We will clarify that we meant observations received *from* the first computation action - indeed, there can be more observations from subsequent computation actions.

---

### Meta-Review · Area_Chair_Fkm7 · 2024-02-02

**Recommendation:** Accept (Poster)
**Confidence:** 4

**Metareview:**

The paper propose an extension to situated planning that allow action to be dispatched before planning terminates.

The main weakness of the paper lies in its somewhat incremental nature, as it naturally extends situated planning and the associated established approaches.

Nevertheless, all reviewers agree that the paper tackles an extension of situated planning that is interesting in itself and relevant in practice. The paper is clearly written and provides a sound theoretical foundation that may be used in further work.
In addition, the authors provide a sensible meta-reasoning framework that, under some simplifying assumptions, can be used alongside a situated planner to the concurrent planning and acting problem of interest.

**Ethical Considerations:**

(1) Not Applicable: The paper does not have any ethical considerations to address